# Preparation and Storage of Cryoprecipitate Derived from Amotosalen and UVA-Treated Apheresis Plasma and Assessment of In Vitro Quality Parameters

**DOI:** 10.3390/pathogens11070805

**Published:** 2022-07-18

**Authors:** Katarina Kovacic Krizanic, Florian Prüller, Konrad Rosskopf, Jean-Marc Payrat, Silke Andresen, Peter Schlenke

**Affiliations:** 1Department of Blood Group Serology and Transfusion Medicine, Medical University Graz, Auenbruggerplatz 48, A-8036 Graz, Austria; katarina.kovacickrizanic@medunigraz.at; 2Clinical Institute of Medical and Chemical Laboratory Diagnostics, Medical University Graz, Auenbruggerplatz 15, A-8036 Graz, Austria; florian.prueller@uniklinikum.kages.at; 3Cerus, Europe B.V., Stationsstraat 79-D, 3811 MH Amersfoort, The Netherlands; jpayrat@cerus.com (J.-M.P.); sandresen@cerus.com (S.A.)

**Keywords:** cryoprecipitate, pathogen reduction, amotosalen, in vitro, fibrinogen, factor VIII, factor XIII, von Willebrand factor, thrombelastography, thrombin generation

## Abstract

Cryoprecipitate is a plasma-derived blood product, enriched for fibrinogen, factor VIII, factor XIII, and von Willebrand factor. Due to infectious risk, the use of cryoprecipitate in Central Europe diminished over the last decades. However, after the introduction of various pathogen-reduction technologies for plasma, cryoprecipitate production in blood centers is a feasible alternative to pharmaceutical fibrinogen concentrate with a high safety profile. In our study, we evaluated the feasibility of the production of twenty-four cryoprecipitate units from pools of two units of apheresis plasma pathogen reduced using amotosalen and ultraviolet light A (UVA) (INTERCEPT^®^ Blood System). The aim was to assess the compliance of the pathogen-reduced cryoprecipitate with the European Directorate for the Quality of Medicines (EDQM) guidelines and the stability of coagulation factors after frozen (≤−25 °C) storage and five-day liquid storage at ambient temperature post-thawing. All pathogen-reduced cryoprecipitate units fulfilled the European requirements for fibrinogen, factor VIII and von Willebrand factor content post-preparation. After five days of liquid storage, content of these factors exceeded the minimum values in the European requirements and the content of other factors was sufficient. Our method of production of cryoprecipitate using pathogen-reduced apheresis plasma in a jumbo bag is feasible and efficient.

## 1. Introduction

Cryoprecipitate (Cryo) is a plasma-derived blood product, enriched for fibrinogen (Fb), factor VIII (FVIII), factor XIII (FXIII) and von Willebrand factor (vWF). The discovery of Cryo in the 1960s and its first indication as a replacement for FVIII has led to great support to hemophilia A patients [1]. Clinical settings where cryoprecipitate has been used have changed over time or have become partially obsolete, but are still numerous. There are reports on uses in surgical bleeding, obstetrics, uremic bleeding, liver diseases and hemostatic abnormalities, acquired hypofibrinogenemia, congenital fibrinogen or FXIII deficiency, Von Willebrand disease or major hemorrhage in trauma [2,3,4]. Recommendations for the use of Cryo have been published in recent years and can differ in several countries [5,6,7]. Due to infectious risk, the use of Cryo in Central Europe diminished over the last decades. However, after the introduction of various pathogen-reduction technologies (PRT) for plasma, Cryo production in blood centers is a feasible alternative to pharmaceutical fibrinogen concentrate with a high margin of safety [8,9,10]. The production independence and self-sufficiency of blood banks could be an important advantage in times of shortage of supply routes in case of pandemic or other national or international crises.

Our aim is to evaluate the feasibility and efficacy of the production of cryoprecipitate from a pool of two units of apheresis plasma pathogen-reduced (PR) using amotosalen and ultraviolet light A (A-UVA, INTERCEPT^®^ Blood System). The main objective of our study is compliance of PR-Cryo with the European Directorate for the Quality of Medicines (EDQM) guidelines (fibrinogen > 140 mg/unit, FVIII > 50 IU/unit, vWF > 100 mg/unit, residual cells in plasma below limits) [11] and stability of coagulation factors after frozen (≤−25 °C) storage and five-day liquid storage at ambient temperature post-thawing. The stability of non-PR cryoprecipitate for up to 5 days post-thawing at ambient temperature has been demonstrated [12]. An expanded product shelf life and storage at ambient temperature would ensure product availability for rapid transfusion and cover the needs of emergency department services.

## 2. Results

The volume of plasma units after PR treatment was 610 ± 8 mL. (mean ± SD). Cryo units after separation were of 90 ± 6 (min 72, max 102) mL.

### 2.1. Coagulation Factor levels

#### 2.1.1. Apheresis Plasma Pre- and Post-Pathogen-Reduction Treatment

The residual cells in plasma before pathogen reduction (Table 1) were within the limits for leucocyte-depleted fresh frozen plasma as stated in the EDQM Quality Guidelines (RBC < 6.0 × 10^9^/L, Leukocytes < 1 × 10^6^/U, Platelets < 50 × 10^9^/L) [11].

The changes in coagulation factor content following PR treatment are presented in Table 1. After PR with A-UVA we observed a mean 13%, 21%, 5% and 4% loss of fibrinogen, FVIII, vWF and FXIII, respectively. Although the reduction is statistically significant, the mean content of FVIII is two-fold the minimum requirement of the EDQM for PR-Plasma, and for fibrinogen we observed only 13% of the allowed 40% loss [11]. The ADAMTS13 activity remained stable.

#### 2.1.2. Cryoprecipitate

All produced Cryo units fulfilled the European criteria for fibrinogen, FVIII and vWF activity, respectively (Fb > 140 mg/unit, FVIII > 50 IU/unit, vWF > 100 IU/unit; FXIII and ADAMTS13 are not included in the requirements). These criteria are designed for PR-Cryo obtained from FFP prepared from one unit of whole blood. In our study, cryoprecipitate units were obtained from apheresis plasma with a larger output volume. Based on our treated plasma volume (650 mL), the expected coagulation factor content in our cryoprecipitate units should exceed the European requirements 2.4-fold.

After 24-months frozen storage, the content and activity of the tested coagulation factors remained stable; the observed changes for Fb, FVIII and vWF were not statistically significant (*p* = 0.899 for Fb, *p* = 0.285 for FVIII and *p* = 0.365 for vWF Ac, respectively). Furthermore, all produced Cryos met the volume-adapted European criteria and exceeded the requirements by 4.5-fold, 4.7-fold and 4.3-fold for fibrinogen, FVIII and vWF activity (Figure 1A–C). The FXIII content and TEG had no significant changes (*p* = 0.547, and *p* = 0.717, respectively) (Figure 1D,F). ADAMTS13 activity and TGA show significant changes (*p* = 0.011, and *p* = 0.047, respectively), but no loss over the observed storage period (Figure 1E,G). The 5-day storage at ambient temperature post-thawing resulted in a mean 14%, 22% and 15% loss of fibrinogen, FVIII and vWF, respectively. The observed reduction in coagulation factor content and activity was statistically significant (*p* < 0.001 for the three parameters (Figure 1A–C). No statistically significant losses were detected for FXIII (although significant change), ADAMTS13, TEG and TGA (Figure 1D–G). The length of the frozen storage had no statistically significant effect on the stability or reduction of coagulation factor content post-thawing.

All Cryo units met the volume-adapted quality criteria and still exceeded the European standards by 3.9-fold, 3.7-fold and 3.6-fold for fibrinogen, FVIII and vWF activity after 24-month frozen storage and 5-day at ambient temperature post-thawing, respectively (Figure 1A–C). FXIII content and ADAMTS13 activity remained stable over the complete observed time period without statistically significant losses (Figure 1D,E). These findings were confirmed by thromboelastography and thrombin generation assays, which show stable coagulation and no significant loss up to 120 h (Figure 1F,G).

## 3. Discussion

The problem of non-pathogen-reduced Cryo is the short period of potential usage as it must be transfused as soon as possible following thawing [11] due to the risk of bacterial contamination. This short life span can be overcome by introducing pathogen reduction of the initial plasma.

We present results from a cryoprecipitate in vitro study, which is novel in its combination of high base value of plasma volume obtained by apheresis (concurrent plasma, leucocyte depleted), pathogen reduction, prolonged PR-Cryo frozen storage period of 24 month and prolonged storage of 120 h at ambient temperature after thawing, respectively. In our hands, despite observing a statistically significant decrease in the Fb, FVIII and vWF content at RT, all produced PR-Cryo units fulfilled the European quality requirements (EDQM) [11] for fibrinogen, FVIII and vWF (Fb > 140 mg/unit, FVIII > 50 IU/unit, vWF > 100 IU/unit) including the data after 5 days of storage at ambient temperature. The content and the activity of the other tested coagulation factors FXIII and ADAMTS13 remained stable over the complete frozen and ambient-temperature storage period, there were no statistically significant losses. The EDQM Quality Criteria are designed for PR cryoprecipitate obtained from fresh frozen plasma prepared from one unit of whole blood. Using a large-input fresh plasma volume (600–650 mL) in a jumbo bag for freezing and thawing, we observed more than three times the amount of the required clotting-factor content in our PR-Cryo products. Based on our treated plasma volume, the expected coagulation factor content in our cryoprecipitate units should exceed the European requirements by 2.4 fold. The coagulation-factor content remained stable over the complete observed frozen storage and the length of the frozen storage did not significantly affect the stability of coagulation factors post-thawing. In concordance with our observations, similar results were obtained by other authors who tested the stability of cryoprecipitate at ambient temperature: 

Sheffield et al. reported no significant loss of coagulation-factor activity for cryoprecipitate stored at ambient temperature for up to 24 h. They even observed a small but statistically significant increase in activity, and tried to explain this fact by slow completion of the dissolution process of the large proteins at ambient temperature [13].

Furthermore, Green et al. were able to prove that non-PR cryoprecipitate, stored at ambient temperature for 72 h, maintains hemostatic properties and meets UK guideline specifications [14]. Moreover, Cushing et al. showed that PR cryoprecipitate could be stored at ambient temperature for up to five days without losing clotting-factor levels or fibrin clot strength [15]. A similar study of Thomas et al. proved the ability of PR-Cryo to rescue dilutional and lytical coagulopathy in comparison with fibrinogen concentrates and non-PR-Cryo in an in vitro study with a storage time of even 10 days at room temperature [16].

Regarding the post-thaw changes of coagulation factors over five days at RT it seems that the results of non-PR Cryo and those of our PR-Cryo differ. Lokhandwala et al. report for their non-PR Cryo products over 120 h at RT a significant reduction of FVIII, whereas Fb and vWF had no significant losses [12]. Similar results are described by Thomson et al. FVIII levels declined significantly after 120 h storage at ambient temperature post-thawing, but Fb and vWF levels were not reduced significantly [17]. This is in contrast to our results of PR-Cryo, where we saw moderate but significant reductions after 120 h at RT in FVIII, Fb and vWF contents, respectively, but the European requirements were fulfilled. Our global coagulation results based on thrombolastography and thrombin generation show stable coagulation and no significant loss up to 120 h, as described in recent literature [16,17].

Residual cells in our products met the requirements for leucocyte-depleted fresh frozen plasma, which contributes to the transfusion safety by reducing the risk of alloimmunization against HLA or other leucocyte antigens and also diminishes the risk of CMV infection and of febrile nonhemolytic transfusion reactions. 

The observed loss in coagulation-factor content following pathogen-reduction treatment is well-known and comparable with the data previously described by different authors [18,19,20,21,22,23]. In a complex proteomic analysis, Kamyszek et al. showed that in Cryo made from A-UVA PR plasma, the pathogen-reduction process does not significantly impact the coagulation-factor content or function of Cryo if compared with non-PR Cryo [24]. Moreover, the pathogen-reduction treatment reduces the risks for transfusion-transmitted infections and enables a prolonged Cryo storage at ambient temperature, as bacteria can otherwise proliferate in a thawed non-PR cryoprecipitate stored at ambient temperature for longer than 4 h [25].

The loss of coagulation factors in the cryoprecipitate production process has also been already described by other authors [8,10]. Cid et al. reported a 35% and 40% decrease in fibrinogen and FVIII content in A-UVA-treated cryoprecipitates, prepared from whole-blood-derived plasma components and stored for 1 month at −30 °C, while the quantity and quality of vWF were retained [10]. Backholer et al. described a 41%, 31% and 26% reduction in fibrinogen activity, FVIII and FXIII content in A-UVA treated cryoprecipitates, respectively [8]. In our hands, the production of PR-Cryo and 1 month storage at below −25 °C resulted in a 54%, 66% and 43% mean reduction of the fibrinogen, FVIII and vWF levels. 

There are certainly limitations to our study, which are mentioned as follows. Because of the high amount of sample draws and the need for sufficient cryoprecipitate volume for testing, we used different small groups (*n* = 6) of PR-Cryo produced from different donors for each frozen-storage time point. Therefore, the statistical comparison between the stored batches is limited and some of the trends may be significant if larger numbers of samples per group were used.

The critical-process steps of precipitation and centrifugation in the Cryo production may lead to an unsatisfactory loss of clotting factors. In our investigation, we focused on the quality of frozen and thawed PR-Cryo with respect to the requirements of the EDQM guidelines. To improve the efficacy of cryoprecipitate production, further efforts have to be put in place. 

In summary, in our hands, an in-house production of PR cryoprecipitate from apheresis plasma that fulfills the European quality standards is feasible and efficient. Due to the sufficient in vitro stability of the measured coagulation-factor content and functional parameters after 24 months frozen storage and 5 days post-thawing, PR-Cryo may provide an alternative product for specific clotting-factor preparations. The longer shelf life addresses in particular the needs of emergency rooms to treat hemorrhagic shock, as early fibrinogen or cryoprecipitate therapy seems to lead to clinical benefit [4,26] and would help to reduce the high wastage rates. 

As an alternative product for fibrinogen concentrate, widely used in Europe, PR-Cryorepresents an important source of fibrinogen and other coagulation factors with a high margin of safety. PR-Cryo using the INTERCEPT^®^ Blood System for plasma can be produced using blood establishment’s equipment and techniques. The cost effectiveness of cryoprecipitates compared to fibrinogen concentrates has been shown [27,28], but the relevance of the economic comparison with our studied PR-Cryo products is limited as their study was focused on non-PR Cryo preparations.

Cryoprecipitate production in blood banks and the local self-supply, independent from the international product availability and transportation conditions, is an important safety aspect that contributes to a continuous supply chain, especially in times of national or international crises.

## 4. Materials and Methods

### 4.1. Plasma Preparation and Pathogen Reduction with Amotosalen/UVA

Plasma units used in our study were obtained as concurrent plasma with a volume of 325 mL as a part of platelet apheresis using the Trima Accel^®^ (Terumo Germany, D-65760 Eschborn, Germany), and the Amicus^®^ (Fresenius Kabi Austria, A-8055 Graz, Austria) device according to our standard operating procedure for platelet and plasma apheresis.

After obtaining written informed consent, 48 concurrent plasma units were collected from volunteer blood donors (34% blood group O and 66% non-O) meeting the Austrian donor criteria for blood donation. A total of 24 plasma pools were prepared each from two ABO-identical plasma units to yield 600 to 650 mL and subsequently pathogen-reduced using amotosalen and UVA-light (INTERCEPT^®^ Blood System for plasma, Plasma-Set INT3104B and INT100 UVA illuminator, Cerus Europe BV., 3811 MH Amersfoort, The Netherlands) within 6 h post-collection. Samples for analysis were drawn from plasma units before PR (after pooling) and after PR, respectively, and stored at <−65 °C until tested after the corresponding frozen-storage period of the PR-Cryo (i.e., 1, 3, 12, 24 month), with the exception of residual cells in plasma, which were tested from unfrozen samples.

Following pathogen reduction, each pooled plasma unit was transferred into a single jumbo bag (INTERCEPT Cryo processing container ICPC, a prototype not approved in the EU, with a volume of 650 mL, Cerus Eurpe BV., 3811 MH Amersfoort, The Netherlands) after sterile connection using a sterile connecting device (TSCD II, Terumo Germany, D-65760 Eschborn) and frozen to <−25 °C core temperature within 45 min using a contact shock freezer (KLF48, CLST, A-9130 Poggersdorf Austria) according to our standard operating procedure (Figure 2A).

Within 1 to 30 days of frozen storage, plasma units were thawed at 2 °C to 6 °C overnight (app. 18 to 24 h) and centrifuged with a water-cooled centrifuge (Roto Silenta 630 RS, Andreas Hettich GmbH, D-78532 Tuttlingen, Germany) using a hard-spin program. The centrifugation settings were 4000× *g*, 12 min 30 s with an acceleration rate of 7 and deceleration rate of 6 at 4 °C. Prior to the plasma centrifugation the centrifuge was cooled down to 4 °C by conducting an empty run. The cryo-poor supernatant was transferred into two of the attached INTERCEPT plasma-storage containers, leaving sufficient plasma in the ICPC bag resulting in 72 to 102 mL of pathogen-reduced cryoprecipitate (PR-Cryo). PR-Cryo was transferred to the last INTERCEPT plasma-storage container, frozen according to the protocol for plasma and stored at <−25 °C (Figure 2B).

### 4.2. Frozen Storage, Thawing and Sampling of PR-Cryo

Subsequently, four series of six PR-Cryo units each were thawed with Plasmatherm Plasma Thawing System (Barkey GmbH, D- 33818 Leopoldshoehe, Germany) at 37 °C after 1 month (±1 week), 3 months (±1 week), 12 months (±2 weeks) and 24 months ± 2 weeks) of frozen storage, and kept at ambient temperature (20–24 °C) over 120 h. Samples were taken 1 h, 24 h, 72 h and 120 h after thawing and were stored at below −65 °C until tested. Within six weeks thereafter, coagulation tests of the PR-Cryo samples took place simultaneously with the corresponding samples from plasma units (before and after PR) after the frozen-storage time periods of 1, 3, 12 and 24 months of each series, respectively. All samples were transported on dry ice to the lab.

### 4.3. Residual Cells

Residual cells were tested in the plasma products prior to pathogen reduction. White blood cells (WBC) and red blood cells (RBC) were quantified using flow cytometry (BD FACSCalibur™ with Trucount™ absolute count tubes, Becton Dickinson Austria GmbH, Wien, Austria). Residual platelets (PLT) were quantified on the ADVIA 2120 Hematology Analyzer (Siemens Healthcare Diagnostics GmbH, Vienna, Austria).

### 4.4. Coagulation Testing

Fibrinogen (Multifibren^®^ U reagent, Siemens Healthcare Diagnostics GmbH, Vienna, Austria), Factor VIII (Pathromtin^®^ SL reagent and factor VIII deficient plasma, Siemens Healthcare Diagnostics GmbH, Vienna, Austria), Factor XIII (Berichrom FXIII reagent Siemens Healthcare Diagnostics GmbH, Vienna, Austria) as well as vWF Activity and ADAMTS13 (Innovance vWF Ac assay, Siemens Healthcare Diagnostics GmbH, Vienna, Austria) were measured on the Atellica COAG 360 Coagulation System (Siemens Healthcare Diagnostics GmbH, Vienna, Austria). Thrombin Generation was measured on the BCS XP Coagulation Analyzer using Endogenous Thrombin Reagent (Innovance ETP, (Siemens Healthcare Diagnostics GmbH, Vienna, Austria). Functional fibrinogen test was performed on the TEG 5000 analyzer (Haemonetics, Vienna, Austria) at the various time points, respectively.

### 4.5. Statistical Analysis

Repeated-measures analysis of variance (ANOVA) model was used to examine the impact of time in ambient temperature (RT) storage on various plasma factors and other quality-parameter test outcomes at each frozen-storage time point. *p*-values are based on Pillai’s Trace test statistics due to its powerfulness and robustness with small sample sizes. Additionally, least-squares (LS) means by Dunnett test were used to examine the impact of time in RT storage as compared to 1 h after thawing.

To evaluate the frozen storage, two-way mixed ANOVA model was applied to check the stability of the PR-Cryo during frozen storage with an assessment of between-subject effect. The ambient-temperature storage effects and frozen-storage effects on the coagulation parameters were tested by multivariate approach using Pillai’s Trace test statistics.

For all statistical comparisons, a two-sided *p*-value < 0.05 was considered statistically significant.

## Figures and Tables

**Figure 1 pathogens-11-00805-f001:**
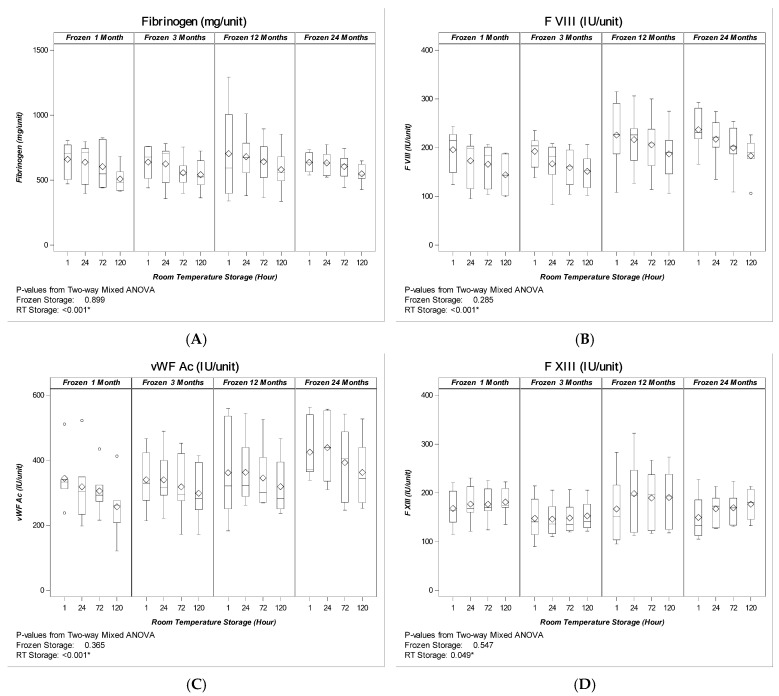
Pathogen-reduced Cryoprecipitate (PR-Cryo). Coagulation factor stability after storage at <−25 °C and up to 5 days at ambient temperature (*n* = 24): (**A**) Fibrinogen, (**B**) Factor VIII, (**C**) vWF, (**D**) Factor XIII, (**E**) ADAMTS13, (**F**) Thromboelastography—Max. Amplitude after 30 min, and (**G**) Thrombin generation assay. (**A**–**C**) are the markers required by the EU regulatory body. (**D**–**G**) are markers not required by the EU regulatory body. “◊” represent mean values. “–” represent median values. “o” represents outliers. *p*-values were calculated from two-way mixed ANOVA model with frozen storage as between-product factor and room temperature (RT) storage as within-product factor. A *p*-value less than 0.05 indicates a significant effect of 24-month frozen storage or 5-day RT storage; significance is indicated by an asterisk (*).

**Figure 2 pathogens-11-00805-f002:**
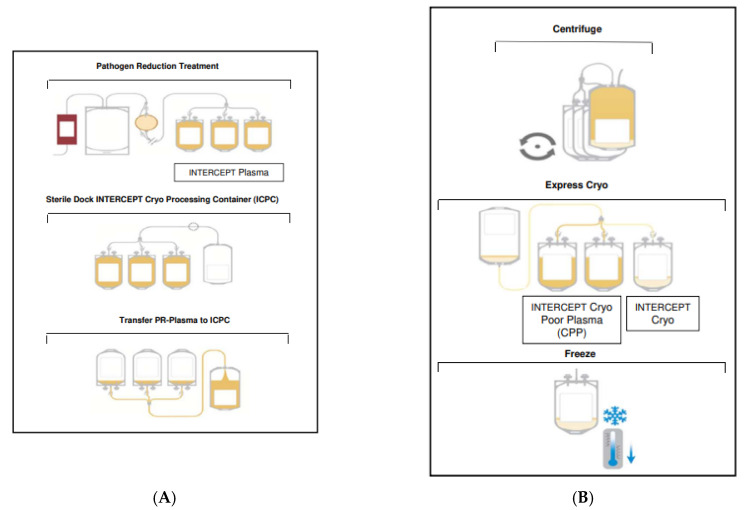
(**A**) Flow chart of plasma pathogen reduction. PR-plasma in the three containers is transferred into the jumbo ICPC bag after sterile connection. Afterwards, the ICPC bag and the three empty containers are shock-frozen together (not illustrated). (**B**) After being thawed and centrifuged the Cryo-poor supernatant is transferred into two of the attached plasma-storage containers; the PR-Cryo is transferred to the last plasma-storage container with subsequent freezing at <−25 °C.

**Table 1 pathogens-11-00805-t001:** The loss due to PR is significant for fibrinogen, FVIII, vWF Ac and FXIII, but not significant for ADAMTS13. A *p*-value less than 0.05 is flagged with an asterisk (*).

Parameter	Pre- Treatment	Post- Treatment	Loss (%)	Significance (*p*)
RBC × 10^6^/L	10 ± 7			
WBC × 10^6^/U	0.1 ± 0.1			
PLT × 10^9^/L	6 ± 3			
Fibrinogen mg/dL	270 ± 40	234 ± 31	13 ± 5	<0.001 *
FVIII IU/dL	128 ± 30	101 ± 21	21 ± 7	<0.001 *
vWF Ac IU/dL	114 ± 30	107 ± 29	5 ± 9	0.008 *
FXIII Ac IU/dL	109 ± 15	104 ± 15	4 ± 3	<0.001 *
ADAMTS13%	93 ± 6	92 ± 7	1 ± 6	0.359

## Data Availability

All data are safely stored and available from the authors who were responsible for the manufacturing and quality control analysis.

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
