# Peer review of "Preparation and Storage of Cryoprecipitate Derived from Amotosalen and UVA-Treated Apheresis Plasma and Assessment of In Vitro Quality Parameters"

_pathogens, 2022, doi:10.3390/pathogens11070805_

Round 1

Reviewer 1 Report

1. Introduction

Line 38 FVIII

2. Results

Table 1. Consider adding one last column for showing (in)significant differences between pre and post-treatment

Figure 1. The significant differences in graphs are not easily visible and some seem to be in the wrong place in the current draft. Consider either marking them clearly on the graph or put the description in the caption.

I suggest making Figure 1 into two Figures. Figure 1 for the markers required by the EU regulatory body and Figure 2 for the graphs of other 4 variables.

Line 64 Maybe the volumes of plasma and cryoprecipitate can be moved to where there is a brief explanation about their larger volumes. We usually see cryo units with volumes of around 10-15 mL (before pooling) and the Cryo of 90 mL was confusing to me in the beginning.

3. Discussion

Consider mentioning the current shelf-life of Cryos post-thawing (6h) to highlight the extended period of testing in this study.

Authors could compare the post-thaw changes in Fibrinogen, factor VIII, and von Willebrand factor levels in their work with the results of regular Cryos (non-PR).

I think it would be also interesting to mention the findings of another recent work on PR-Cryos by Thomas KA et al. 2021 “Effects of pathogen reduction technology and storage duration on the ability of cryoprecipitate to rescue induced coagulopathies in vitro” where they discuss their own findings.

Author Response

Dear reviewer 1,

Thank you very much for reviewing our manuscript. Please find our answers in the enclosed attachment.

Kind regards

Konrad Rosskopf for all authors

Reviewer 2 Report

This is a nice small in-house study showing that “production of PR cryoprecipitate from apheresis plasma that fulfills the European quality standards is feasible and efficient.” As PR cryo is seldomly used in many countries, including the US, this study adds to the current body of literature in this area that may be of interest to readers. However, there are two issues that hindered enthusiasm for this manuscript.

1) As the authors describe, there have many multiple other studies drawing similar conclusions as those drawn here. In particular, References 8, 10, 15, and 16 have done similar studies and have shown similar results. The authors should provide data or an explanation of the novelty of their study.

2) Figure 1: It appears that some of the frozen units (i.e. 1 month storage) have reduced fibrinogen, FVIII, and vWF over 5 days, but other frozen units (i.e. 24 months frozen for vWF) do not decrease over 5 days. The text states that frozen storage time did not have a significant effect on coagulation factor decreases over 5 days, but that does not appear to be the case. Please explain.

Table 1: When was post-treatment plasma evaluated (i.e. time between pre and post treatment)?

Line 92: Add “, respectively” after “vWF activity”.

Also, some of the trends may be significant if larger numbers of samples per group were used. Please add this to the Limitations in the Discussion.

Figure 1: There appear to be * indicating statistical significance, but it is not clear which groups are being compared. Please clarify.

Line 106: Please explain how the thromboelastography and thrombin generation assays confirm the other findings.

Figure 2A: Please add labels as was done for figure 2B.

Line 191: Leukocyte-depletion also diminishes CMV infection and febrile nonhemolytic transfusion reactions. Please add this comment.

Line 236: Please clarify the phrase “with their limitations of Cryos preparations being non PR”.

Author Response

Dear reviewer 2,

Thank you very much for reviewing our manuscript. Please find our answers in the enclosed attachment.

Kind regards

Konrad Rosskopf for all authors
